# Synthesis, Cytotoxicity and Molecular Docking of New Hybrid Compounds by Combination of Curcumin with Oleanolic Acid

**DOI:** 10.3390/biomedicines11061506

**Published:** 2023-05-23

**Authors:** Katarzyna Sowa-Kasprzak, Ewa Totoń, Jacek Kujawski, Dorota Olender, Natalia Lisiak, Lucjusz Zaprutko, Błażej Rubiś, Mariusz Kaczmarek, Anna Pawełczyk

**Affiliations:** 1Chair and Department of Organic Chemistry, Poznan University of Medical Sciences, Grunwaldzka 6 Str., 60-780 Poznań, Poland; 2Department of Clinical Chemistry and Molecular Diagnostics, Poznan University of Medical Sciences, Rokietnicka 3 Str., 60-806 Poznań, Poland; etoton@ump.edu.pl (E.T.);; 3Department of Cancer Diagnostics and Immunology, Gene Therapy Unit, Greater Poland Cancer Centre, Garbary 15 Str., 61-866 Poznań, Poland

**Keywords:** curcumin, oleanolic acid, hybrid compounds, cytotoxic activity, in silico studies

## Abstract

Curcumin and oleanolic acid are natural compounds with high potential in medicinal chemistry. These products have been widely studied for their pharmacological properties and have been structurally modified to improve their bioavailability and therapeutic value. In the present study, we discuss how these compounds are utilized to develop bioactive hybrid compounds that are intended to target cancer cells. Using a bifunctional linker, succinic acid, to combine curcumin and triterpenoic oleanolic acid, several hybrid compounds were prepared. Their cytotoxicity against different cancer cell lines was evaluated and compared with the activity of curcumin (the IC50 value (24 h), for MCF7, HeLaWT and HT-29 cancer cells for **KS5**, **KS6** and **KS8** compounds was in the range of 20.6–94.4 µM, in comparison to curcumin 15.6–57.2 µM). Additionally, in silico studies were also performed. The computations determined the activity of the tested compounds towards proteins selected due to their similar binding modes and the nature of hydrogen bonds formed within the cavity of ligand−protein complexes. Overall, the curcumin-triterpene hybrids represent an important class of compounds for the development of effective anticancer agents also without the diketone moiety in the curcumin molecule. Moreover, some structural modifications in keto-enol moiety have led to obtaining more information about different chemical and biological activities. Results obtained may be of interest for further research into combinations of curcumin and oleanolic acid derivatives.

## 1. Introduction

The contemporary strategy to fight cancer is a consequence of the rapid development of knowledge about the molecular basis of cancerous diseases [1,2]. Current anticancer drugs have multidirectional mechanisms of action. They inhibit cell division, cause DNA damage, block transcription and translation, contribute to the induction of tumor cell apoptosis in the final stage and the inhibition of pro-angiogenic factors and their receptors [3]. Many new anticancer drugs targeting angiogenesis are identified in the literature. The results of the in vitro and in vivo evaluation of these drugs show that, apart from inhibiting angiogenesis, they also affect cancer cell proliferation and tumor growth [4].

A number of limitations must be overcome for the treatment of cancer, such as the unselective targeting of cells, multi-drug resistance, relapse of the cancer and poor outcomes. Chemotherapeutic agents target rapidly dividing cancer cells to suppress tumor progression; however, their non-specific cytotoxicity often leads to significant side effects, e.g., neurotoxicity, gastrointestinal toxicity, hematological toxicity, cardiotoxicity, hepatotoxicity and nephrotoxicity [5]. These limitations force scientists to constantly search for new substances and modify existing compounds in order to use them in targeted anticancer therapies [6].

Medicinal plants have been an excellent source of pharmaceutical agents for a long time. About 60% of newly approved small molecule drugs were derived from natural compounds of their derivatives from 1981 to 2019 [7]. The development of integrated research systems and advanced screening procedures for plant bioactive components has ushered in a new era of phytochemical discoveries for the prevention and treatment of complex diseases such as cancer. Bioactive compounds such as berberine, curcumin, crocetin, colchicine, gingerol, lycopene, kaempferol, resveratrol, vincristine, and vinblastine have demonstrated remarkable anticancer potential [8,9].

Curcumin is the most important component of the rhizomes of *Curcuma longa* and was extracted from the turmeric plant in a pure crystalline form for the first time in 1870; the feruloylmethane skeleton of curcumin was confirmed and synthesized by Polish chemists: Kostanecki, Miłobędzka and Lampe in 1910 [10,11]. A wide variety of cellular properties of curcumin have been demonstrated, including antiproliferative [12], pro-apoptotic, anticancer [13], antioxidant, anti-inflammatory [14] and antibacterial activities [15]. The main mechanisms of action by which curcumin exhibits its unique anticancer activity include inducing apoptosis and inhibiting proliferation and invasion of tumors by suppressing a variety of cellular signaling pathways. Several studies reported curcumin’s antitumor activity on breast cancer, lung cancer, head and neck squamous cell carcinoma, prostate cancer, and brain tumors, showing its capability to target multiple cancer cell lines [13]. The anti-inflammatory effect of curcumin is based on its ability to inhibit COX-2, lipoxygenase (LOX), inducible nitric oxide synthase (iNOS), arachidonic acid metabolism, cytokines (interleukins), and tumor necrosis factor NF-κB and the release of steroid hormones [14]. The antioxidant property is attributed to the presence of various functional groups, including methoxy, phenoxy, and carbon–carbon double bonds in its structure. Curcumin can inhibit oxidative damage caused by stress in the brain, liver and kidneys of rats [16]. Lipid peroxidation is significantly reduced in rats treated with curcumin before applying γ-radiation [17]. Curcumin increases enzymatic antioxidant activity by increasing the expression of methionine sulfoxide reductase (MSRA) and increasing the levels of the enzymes MSRA, SOD, CAT and GPx [18], and may act as an antioxidant against oxidative stress in rats with diabetes mellitus by increased SOD expression in cochlear fibroblasts [19]. This compound can be considered as a promising antibacterial agent, but with very selective activity towards individual species and strains. Adamczak et al. tested its efficacy against over 100 strains of pathogens belonging to 19 species and confirmed the much greater sensitivity of Gram-positive than Gram-negative bacteria. Curcumin exhibited strong antibacterial properties in the in vitro tests, and it was very effective against some species and strains: *Streptococcus pyogenes*, methicillin-sensitive *S. aureus*, *Acinetobacter lwoffii*, and individual strains of *Enterococcus faecalis* and *Pseudomonas aeruginosa* [15].

Oleanolic acid (OA) is a pentacyclic triterpenoid isolated from a plethora of plants and herbs. In the last decade, the major international databases have recorded several thousand publications dealing with oleanolic acid, reflecting the great interest of the scientific community [20]. Apart from its ecological roles in plants, some pharmacological activities such as antitumor, anti-inflammatory, antioxidant, anti-diabetic, and antimicrobial effects have been attributed to oleanolic acid in different models of diseases [21]. OA has been found to be active at various stages of tumor development, including the inhibition of tumor promotion, invasion and metastasis; its cytotoxic effects have been reported on several cancer lines, suggesting its antitumor activity [22]. In vitro studies have shown several mechanisms of the anti-inflammatory effect of triterpenes. Their activity consists mainly in inhibiting the activity of enzymes involved in the inflammatory reaction, such as phospholipase A_2_, cyclooxygenase, lipoxygenase, nitric oxide synthase and elastase. They can also reduce the production of prostaglandins and pro-inflammatory cytokines: TNF-α, IFN-γ, IL. In addition, the inhibition of the transcription factor NF-kB, which plays a role in inflammatory processes and tumor progression, was observed [23]. Pentacyclic triterpenes have shown promising results in studies on animal models of diabetes. The observed hypoglycemic and hypolipemic effects are related to the stimulation of insulin secretion by oleanolic acid [24]. Oleanolic acid and other triterpenes also exhibit a broad spectrum of antimicrobial activity, mainly against Gram-positive bacteria. There have been reports of research into prototype drugs that can be derived from triterpenes that can be used against infections caused by drug-resistant bacterial species [25].

By taking into account the knowledge about the pharmacological, structural, and molecular interaction profiles of anticancer drugs, hybrid molecules, called molecular consortia in general [26], are designed by chemical hybridization, wherein two or more drugs having different activities can be co-formulated by connection through a stable or metabolizable linker, allowing simultaneous delivery. Hybrid molecules based on natural products can be generated naturally or synthetically by combining entire or partial natural scaffolds [27].

Curcumin has been employed as a promising lead compound for structural modification to develop novel anticancer agents by medicinal chemists worldwide [28]. Literature [29] about the creation of analogs of curcumin affecting its better bioavailability due to the improvement of physicochemical properties and improving biological activity includes: modifications within the alkyl unit between the aromatic rings, especially in the central diketone/enol moiety (at the C-3, C-4, C-5 position), modifications using double bonds (C-1, C-6), the introduction or exchange of substituents into the aromatic rings and the formation of complexes of curcumin and its derivatives with metal ions. A very important modification of the structure of curcumin is the use of reactive phenolic groups at the C-4′ positions, for example, by its esterification (Figure 1). There is also a lot of literature [7] on the naturally abundant OA that served as scaffolds for chemical modifications to improve their aqueous solubility, bioavailability and selectivity, and therapeutic potential. These structural modifications were principally introduced at the C-3, C-12 and C-28 positions of the oleanolic acid scaffold (Figure 1).

Previously, we reported compounds containing curcumin or oleanolic acid skeleton with selected NSAID derivatives [30,31]. Conjugation with anti-inflammatory drugs is justified by the fact that hybridization is considered a leading synthetic trend in medicinal chemistry strategy on the one side, and the inflammatory background of many types of cancer treatment, such as hepatocellular carcinoma (HCC) on the other. We also synthesized and described the linked triterpene-phenolic hybrids and their molecular parameters, and the preliminary anti-inflammatory and antinociceptive activity characterizing these molecules as potential drugs were calculated and predicted [32]. In the literature, we find several pyrimidino-, pyridazino- and curcumin-steroid derivatives; cytotoxic effects versus human breast cancer cells (MCF7) [33] were shown; and potent antibacterial hybrid compounds containing oleanolic acid and curcumin [34].

In this work, we synthesized several derivatives based on curcumin or its heterocyclic analogs (with the pyrazole or isoxazole system instead of the curcumin central diketone system) and oleanolic acid using succinic acid moiety as a bifunctional linker (Figure 2). In the next stage of the research, the obtained hybrid individuals were assessed for their cytotoxicity effect against HeLaWT, HT-29 and MCF7 cells.

## 2. Materials and Methods

### 2.1. Chemical Synthesis

#### 2.1.1. General

All commercially available solvents and reagents used in our experiments were graded “pure for analysis” (Aldrich^®^, Germany, Fluka^®^, USA, Chempur^®^, Poland and POCh^®^, Poland). The solvents were dried according to the usual procedures. Products were purified by column chromatography using 70–230 mesh silica gel (Merck, Germany). The progress of reactions and purity of products were checked using the TLC method on silica gel plates (DC-Alufolien Kieselgel 60 F_254_, Merck, Germany). The ^1^H- and ^13^C-NMR spectra were recorded using a Bruker Advance 600 MHz spectrometer. Chemical shifts (δ) were expressed in parts per million (ppm), relative to tetramethylsilane (TMS) as an internal standard, using CDCl_3_ as a solvent. Coupling constants (*J*) are expressed in Hertz (Hz). Signals are labeled as follows: s, singlet; d, doublet; dd, double doublet; t, triplet; and m, multiplet. MS spectra were recorded on electrospray ionization mode (ESI-MS) (QTOF mass spectrometer—Impact HD, Bruker, USA).

#### 2.1.2. Synthesis of Compounds


*Oleanolic Acid Methyl Ester; Methyl 3β-hydroxyolean-12-en-28-oate*


Synthesis of oleanolic acid methyl ester from oleanolic acid was performed according to the literature protocol [22]. Yield: 96%; m.p. and spectral data agreed with the literature [22].


*Mono-Oleanoyl hydrogen succinate (**KS1**),*
*4-{[(3*
*β*
*)-28-hydroksy-28-oxoolean-12-en-3-yl]oxy}-4-oxobutanoic acid*


Oleanolic acid (0.94 g, 2 mmol) and DMAP (2.44 g, 20 mmol) were added to a solution of succinic anhydride (2.0 g, 20 mmol) in dry pyridine (10 mL), and the mixture was heated under reflux for 8 h. The solution was left overnight and poured into water (100 mL). The obtained precipitate was filtered off, washed with 5% HCl, dried and crystallized from ethanol. Yield: 90%; m.p. and spectral data agreed with the literature [35].


*M*
*ono-oleanoyl hydrogen succinate methyl ester (**KS1a**); 4-{[(3*
*β*
*)-28-Methoxy-28-oxoolean-12-en-3-yl]oxy}-4-oxobutanoic acid*


Oleanolic acid methyl ester (0.91 g, 2 mmol) and DMAP (2.44 g, 20 mmol) were added to a solution of succinic anhydride (2.0 g, 20 mmol) in dry pyridine (10 mL), and the mixture was heated under reflux for 8 h. The solution was left overnight and poured into water (100 mL). The obtained precipitate was filtered off, washed with 5% HCl, dried and crystallized from ethanol. Yield: 90%; m.p. and spectral data of compound **KS1a** agreed with the literature [35].


*Curcumin isoxazole (**KS2a**); 3,5-Bis[β-(4-hydroxy-3-methoxyphenyl)-ethenyl]isoxazole*


A mixture of curcumin (1 mmol) and hydroxylamine hydrochloride (1 mmol) was refluxed at about 80 °C in ethanol (5 mL) for 16 h. TLC monitored the progress of the reaction. After completion, water and ethyl acetate were added to the reaction mixture, and the organic layer was washed with brine and dried over anhydrous MgSO_4_. After removal of the drying agent and solvent evaporation, the crude product was purified with column chromatography on silica gel using a chloroform/methanol (9:1, *v:v*) mixture as eluent. The product was obtained as an orange-yellow solid. Yield: 78%; m.p. and spectral data of compound **KS2a** agreed with the literature [36].


*Curcumin pyrazole (**KS2b**); 3,5-Bis[*
*β-(4-hydroxy-3-methoxyphenyl)-ethenyl]-1H-pyrazole*


A mixture of curcumin (1 mmol) and hydrazine dihydrochloride (1 mmol) was refluxed at about 80 °C in ethanol (10 mL) for 48 h. TLC monitored the progress of the reaction. After completion, the reaction mixture was cooled to room temperature. The precipitate was filtered under reduced pressure and dried. The crude product was purified with column chromatography on silica gel using a chloroform/methanol (9:1, *v:v*) mixture as eluent. The product was obtained as a red solid. Yield: 79%. The m.p. and spectral data of compound **KS2b** agreed with the literature [36].


*General procedure for the synthesis of compounds **KS3** and **KS4***


Curcumin (0.5 mmol) and DCC (0.8 mmol) were dissolved in chloroform (20 mL), and then **KS1** or **KS1a** (0.6 mmol) and DMAP (0.4 mmol) were added. The reaction mixture was stirred at room temperature for 24 h. TLC monitored the reaction. After completion, hexane (8 mL) was added, and the precipitated dicyclohexylurea was filtered. The organic phase was washed with 5% HCl, 5% NaHCO_3,_ and water and dried over anhydrous MgSO_4_. After removal of the drying agent and solvent evaporation, the crude product was purified with column chromatography on silica gel using a chloroform/methanol (20:1, *v:v*) mixture as eluent.


*Curcumin mono-oleanoyl hydrogen succinate methyl ester (**KS3**)*


The product was obtained as a yellow solid. Yield: 62%, R_f_ = 0.67 (chloroform/methanol 20:1); ^1^H-NMR: *δ* 15.84 (bs, 1H); 9.94 (s, 1H); 7.62 (d, *J* = 15.8 Hz, 2H); 7.16 (dd, *J* = 10.8, 9.3 Hz, 4H); 7.11 (d, *J* = 1.5 Hz, 1H); 7.07 (d, *J* = 3.6 Hz, 1H); 6.56 (d, *J* = 15.8 Hz, 2H); 5.85 (s, 1H); 5.28 (t, *J* = 3.7 Hz, 1H); 4.55 (m, 1H); 3.86 (s, 3H); 3.72 (s, 3H); 3.62 (s, 3H); 2.91 (dd, *J* = 7.0, 5.6 Hz, 1H); 2.75–2.72 (m, 4H); 2.01–1.97 (m, 1H); 1.90–1.86 (m, 2H); 1.72–1.20 (m, 16H); 1.13 (s, 3H); 1.06–1.00 (m, 2H); 0.93 (s, 3H); 0.91 (s, 3H); 0.89 (s, 3H); 0.86 (s, 3H); 0.85 (s, 3H); 0.82–0.76 (m, 1H); 0.72 (s, 3H); ^13^C-NMR: *δ* 183.08; 181.58; 178.28; 171.66; 170.27; 151.36; 150.28; 143.81; 141.26; 139.96; 133.96; 124.26; 123.24; 122.25; 121.05; 111.48; 101.66; 81.49; 55.98; 55.31; 51.50; 47.54; 46.71; 45.84; 41.63; 41.29; 39.28; 38.08; 37.75; 36.92; 33.81; 33.09; 32.58; 32.36; 30.68; 29.54; 29.07; 28.06; 27.67; 25.89; 23.63; 23.49; 23.06; 18.20; 16.82; 15.34; ESI-MS, *m/z*: 922.1643 [M+H]^+^ (C_56_H_72_O_11_).


*Curcumin mono-oleanoyl hydrogen succinate (**KS4**)*


The product was obtained as a yellow solid. Yield: 60%, Rf = 0.74 (chloroform/methanol 20:1); ^1^H-NMR: δ 15.86 (bs, 1H); 9.93 (s, 1H); 7.61 (d, *J* = 15.3 Hz, 1H); 7.54 (d, *J* = 15.8 Hz, 1H); 7.15–7.11 (m, 4H); 7.06 (d, *J* = 8.1 Hz, 1H); 6.93 (d, *J* = 7.6 Hz, 1H); 6.56 (d, *J* = 15.3 Hz, 1H); 6.49 (d, *J* = 15.7 Hz, 1H); 5.85 (s, 1H); 5.28 (t, *J* = 3.7 Hz, 1H); 4.54 (m, 1H); 3.94 (s, 3H); 3.85 (s, 3H); 2.93 (dd, *J* = 15.9; 8.9 Hz, 1H); 2.75–2.73 (m, 4H); 2.04–1.97 (m, 1H); 1.89–1.84 (m, 2H); 1.70–1.20 (m, 16H); 1.14 (s, 3H); 1.07–1.00 (m, 2H); 0.93 (s, 3H); 0.92 (s, 3H); 0.91 (s, 3H); 0.87 (s, 3H); 0.86 (s, 3H); 0.85–0.81 (m, 1H); 0.76 (s, 3H); ^13^C-NMR: δ 184.49; 181.83; 172.92; 171.71; 170.30; 151.37; 148.03; 146.84; 144.63; 141.26; 139.97; 133.98; 124.27; 123.28; 122.38; 121.89; 121.07; 114.88; 111.49; 101.80; 81.48; 55.94; 55.34; 55.24; 48.36; 47.55; 45.71; 41.78; 41.25; 39.44; 38.14; 37.77; 36.92; 33.64; 33.00; 32.55; 31.44; 30.66; 29.55; 29.08; 28.09; 27.46; 25.83; 23.62; 23.51; 22.97; 18.19; 17.16; 16.77; 15.34; ESI-MS, *m/z*: 908.1443 [M+H]^+^ (C_55_H_70_O_11_).


*General procedure for the synthesis of compounds **KS5** and **KS7***


Curcumin isoxazole (**KS2a**, 0.5 mmol) and DCC (0.8 mmol) were dissolved in dioxane (20 mL), and then **KS1** or **KS1a** (0.6 mmol) and DMAP (0.4 mmol) were added. The reaction mixture was stirred at room temperature for 24 h. TLC monitored the reaction. After completion, hexane (8 mL) was added and the precipitated dicyclohexylurea was filtered. The organic phase was washed with 5% HCl, 5% NaHCO_3_ and water, and dried over anhydrous MgSO_4_. After removal of the drying agent and solvent evaporation, the crude product was purified with column chromatography on silica gel using a chloroform/methanol (20:1, *v:v*) mixture as eluent.


*Curcumin isoxazole mono-oleanoyl hydrogen succinate (**KS5**)*


The product was obtained from **KS2a** and **KS1** as a yellow solid. Yield: 61%, Rf = 0.86 (chloroform/methanol 9:1); ^1^H-NMR: δ 9.94 (s, 1H); 7.61 (d, *J* = 15.7 Hz, 1H); 7.32 (d, *J* = 16.4 Hz, 1H); 7.15–7.03 (m, 4H); 6.90 (d, *J* = 16.2 Hz, 2H); 6.56 (d, *J* = 15.7 Hz, 2H); 6.48 (s, 1H); 5.28 (t, *J* = 3.7 Hz, 1H); 4.60–4.48 (m, 1H); 3.87 (s, 3H); 3.86 (s, 3H); 2.95 (dd, *J* = 15.6; 9.3 Hz, 1H); 2.85–2.65 (m, 4H); 2.07–1.91 (m, 1H); 1.84–1.80 (m, 2H); 1.67–1.20 (m, 16H); 1.14 (s, 3H); 1.09–1.04 (m, 2H); 0.93 (s, 3H); 0.92 (s, 3H); 0.88 (s, 3H); 0.86 (s, 3H); 0.84 (s, 3H); 0.83–0.79 (m, 1H); 0.76 (s, 3H); ^13^C-NMR: δ 176.29; 172.87; 171.68; 168.06; 161.82; 154.00; 151.30; 144.60; 143.60; 143.19; 141.22; 140.47; 135.08; 124.24; 123.24; 122.91; 121.85; 121.04; 116.32; 111.45; 110.63; 110.14; 98.59; 81.43; 55.90; 55.61; 55.30; 48.32; 47.51; 45.67; 41.74; 41.21; 39.40; 38.10; 37.73; 36.87; 33.59; 32.96; 32.59; 31.40; 30.62; 29.53; 29.04; 28.05; 27.42; 25.80; 24.62; 23.58; 22.93; 18.15; 17.12; 16.73; 15.29; ESI-MS, *m/z*: 905.1465 [M+H]^+^ (C_53_H_71_NO_9_).


*Curcumin isoxazole mono-oleanoyl hydrogen succinate methyl ester (**KS7**)*


The product was obtained from **KS2a** and **KS1a** as a yellow solid. Yield: 51%, Rf = 0.72 (chloroform/methanol 15:1); ^1^H-NMR: δ 9.83 (s, 1H); 7.61 (d, *J* = 7.5 Hz, 1H); 7.59 (d, *J* = 7.5 Hz, 1H); 7.13–7.06 (m, 4H); 6.94 (d, *J* = 8.2 Hz, 2H); 6.54 (d, *J* = 9.1 Hz, 1H); 6.49 (d, *J* = 15.8 Hz, 1H); 5.83 (s, 1H); 5.28 (t, J = 3.7 Hz, 1H); 4.58–4.50 (m, 1H); 3.95 (s, 3H); 3.86 (s, 3H); 3.68 (s, 3H); 2.96–2.91 (m, 4H); 2.86 (dd, *J* = 13.8; 4.1 Hz, 1H); 2.77–2.70 (m, 1H); 2.00–1.75 (m, 2H); 1.72–1.49 (m, 16H); 1.13 (s, 3H); 1.07–1.00 (m, 2H); 0.93 (s, 3H); 0.92 (s, 3H); 0.88 (s, 3H); 0.86 (s, 3H); 0.85 (s, 3H); 0.83–0.80 (m, 1H); 0.72 (s, 3H); ^13^C-NMR: δ 178.28; 173.25; 171.61; 170.29; 163.29; 154.03; 151.34; 147.99; 146.80; 143.82; 141.10; 139.40; 134.01; 128.46; 127.68; 124.27; 122.24; 121.79; 120.94; 114.85; 111.45; 109.65; 101.49; 81.63; 58.51; 56.01; 55.32; 51.51; 49.80; 47.55; 46.72; 45.85; 41.63; 41.29; 39.28; 38.08; 37.76 36.92; 33.84; 33.09; 32.58; 30.76; 30.68; 29.87; 28.06; 27.68; 26.21; 25.90; 24.64; 23.62; 18.43; 18.19; 16.82; 15.33; ESI-MS, *m/z*: 919.1613 [M+H]^+^ (C_56_H_71_NO_10_).


*General procedure for the synthesis of compounds **KS6** and **KS8***


Curcumin pyrazole (**KS2b**, 0.5 mmol) and DCC (0.8 mmol) were dissolved in dioxane (20 mL) and then **KS1** or **KS1a** (0.6 mmol) and DMAP (0.4 mmol) were added. The reaction mixture was stirred at room temperature for 24 h. TLC monitored the reaction. After completion, hexane (8 mL) was added, and the precipitated dicyclohexylurea was filtered. The organic phase was washed with 5% HCl, 5% NaHCO_3,_ and water and dried over anhydrous MgSO_4_. After removal of the drying agent and solvent evaporation, the crude product was purified with column chromatography on silica gel using a chloroform/methanol (20:1, *v:v*) mixture as eluent.


*Curcumin pyrazole mono-oleanoyl hydrogen succinate (**KS6**)*


The product was obtained from **KS2b** and **KS1a** as a yellow solid. Yield: 61%, Rf = 0.86 (chloroform/methanol 25:2); ^1^H-NMR: δ 9.95 (s, 1H); 7.85 (d, *J* = 8.4 Hz, 1H); 7.83 (d, *J* = 9.0 Hz, 1H); 7.17–6.99 (m, 4H); 6.97 (d, *J* = 16.3 Hz, 2H); 6.87 (d, *J* = 7.7 Hz, 1H); 6.82 (d, *J* = 15.8 Hz, 1H); 6.56 (s, 1H); 5.29 (t, J = 3.7 Hz, 1H); 4.59–4.47 (m, 1H); 3.87 (s, 3H); 3.69 (s, 3H); 2.97–2.78 (m, 4H); 2.83 (dd, *J* = 13.8; 4.1 Hz, 1H); 2.77–2.70 (m, 1H); 2.05–1.73 (m, 2H); 1.68–1.40 (m, 16H); 1.39–1.19 (m, 2H); 1.14 (s, 3H); 0.93 (s, 3H); 0.92 (s, 3H); 0.91 (s, 3H); 0.88 (s, 3H); 0.86 (s, 3H); 0.80 (s, 3H); ^13^C-NMR: δ 176.81; 173.02; 171.61; 153.84; 153.40; 148.87; 148.35; 147.10; 143.60; 141.45; 138.20; 137.90; 136.90; 129.80; 127.70; 124.60; 122.21; 121.85; 120.70; 113.80; 111.45; 110.90; 99.00; 81.69; 56.16; 55.52; 55.22; 49.80; 48.35; 47.54; 45.71; 41.77; 41.24; 39.43; 38.13; 37.77; 36.91; 33.63; 32.99; 32.62; 31.43; 30.66; 29.69; 28.08; 27.45; 26.22; 25.83; 24.65; 18.18; 17.15; 16.73; 15.31; ESI-MS, *m/z*: 904.1573 [M+H]^+^ (C_55_H_70_N_2_O_9_).


*Curcumin pyrazole mono-oleanoyl hydrogen succinate methyl ester (**KS8**)*


The product was obtained from **KS2b** and **KS1a** as a yellow solid. Yield: 58%, Rf = 0.86 (chloroform/methanol 9:1); ^1^H-NMR: δ 9.93 (s, 1H); 7.80 (d, *J* = 8.4 Hz, 1H); 7.78 (d, *J* = 8.9 Hz, 1H); 7.17–7.02 (m, 4H); 6.93 (d, *J* = 16.2 Hz, 2H); 6.87 (d, *J* = 7.5 Hz, 1H); 6.83 (d, *J* = 15.8 Hz, 1H); 6.54 (s, 1H); 5.29 (t, J = 3.7 Hz, 1H); 4.56–4.45 (m, 1H); 3.88 (s, 3H); 3.69 (s, 3H); 2.98–2.75 (m, 4H); 2.83 (dd, *J* = 13.8; 4.1 Hz, 1H); 2.79–2.70 (m, 1H); 2.05–1.71 (m, 2H); 1.68–1.41 (m, 16H); 1.37–1.19 (m, 2H); 1.13 (s, 3H); 0.93 (s, 3H); 0.92 (s, 3H); 0.91 (s, 3H); 0.88 (s, 3H); 0.86 (s, 3H); 0.78 (s, 3H); ^13^C-NMR: δ 178.16; 173.10; 171.60; 153.93; 151.80; 148.80; 148.30; 147.05; 143.53; 141.16; 138.20; 137.90; 136.91; 129.80; 127.24; 122.60; 122.38; 121.80; 121.04; 115.10; 111.52; 109.87; 109.73; 98.90; 81.89; 55.90; 55.67; 55.31; 51.51; 49.80; 47.54; 46.72; 41.66; 41.27; 39.27; 38.08; 37.77; 36.91; 33.86; 33.08; 32.53; 32.35; 30.65; 29.86; 28.05; 27.66; 26.20; 25.92; 23.62; 23.52; 23.04; 18.18; 16.81; 16.69; 15.35; ESI-MS, *m/z*: 918.1866 [M+H]^+^ (C_56_H_72_N_2_O_9_).

### 2.2. Biological Activity Assays

#### 2.2.1. Cell Culture

Three human cancer cell lines and one non-tumorigenic mammary epithelial cell line were used in this study. All cell lines were purchased from the American Type Cell Culture Collection (ATCC, Manassas, VA, USA). The breast adenocarcinoma cell line MCF7 (ATCC^®^ HTB-22™) was maintained as a monolayer in a complete growing medium RPMI-1640 (Biowest, Nuaillé, France), supplemented with 10% (*v/v*) of fetal bovine serum (FBS) (Sigma-Aldrich, Burghausen, Germany). The colorectal adenocarcinoma cell line HT-29 (ATCC^®^ HTB-38™) was cultured in McCoy’s 5A medium (Biowest, Bradenton, FL, USA) with 10% (*v/v*) FBS (Sigma-Aldrich, Burghausen, Germany) addition. The human cervical carcinoma HeLaWT (ATCC^®^ CCL-2TM) was maintained in Dulbecco’s Modified Eagle Medium (DMEM) with 10% (*v/v*) FBS (Sigma-Aldrich, Burghausen, Germany) addition. The epithelial cell line MCF-12A (ATCC^®^ CRL-10782™) was maintained in DMEM-F12 medium (Biowest, Bradenton, FL, USA) supplemented with hydrocortisone (0.5 µg/mL), insulin (10.0 µg/mL), human epidermal growth factor (hEGF) (20.0 ng/mL), cholera toxin (0.1 µg/mL), and 5% (*v/v*) horse serum (all purchased from Sigma-Aldrich, Schnelldorf, Germany).

All cell lines were grown near confluence in 100 × 15 mm cell culture Falcon^®^ Petri dishes (Corning, Warsaw, Poland) at 37 °C in an atmosphere containing 5% (*v/v*) of CO_2_ at 95% (*v/v*) of relative humidity. The absence of mycoplasma was checked routinely using the Mycoplasma Stain Kit (Sigma-Aldrich, Schnelldorf, Germany).

#### 2.2.2. Cell Viability Assay

The cytotoxicity of the studied compounds was assessed on a broad spectrum of mentioned cell lines using the MTT test. The assay was performed as previously described [34]. Briefly, a total of 5 × 10^3^ cells were seeded into each well of 96-well plates in a total medium volume of 100 μL/well. The cells were treated with various concentrations (0.5–100 µM) of **KS1**, **KS2**, **KS3**, **KS4**, **KS5**, **KS6**, **KS7**, **KS8**, and 0.25% DMSO (control group) or Doxorubicin (positive control). All tested compounds were dissolved in DMSO with the final solvent concentration of 0.25% (*v/v*), which did not affect cell viability. Cells were incubated with studied compounds for 24 h, 48 h and 72 h. Subsequently, 10 μL of MTT solution (5.0 mg/mL) (Sigma–Aldrich, Schnelldorf, Germany) was added to each well. The cells were incubated at 37 °C for 4 h, followed by 100 μL of solubilization buffer (10% SDS in 0.01 M HCl) addition. Finally, the absorbance at 570 nm was measured using a Microplate Reader Multiscan FC (Thermo Scientific, Waltham, MA, USA), with a reference wavelength of 630 nm. Relative cell viability was determined using the following formula: Relative cell viability = (mean A570 of experimental groups/mean A570 of control groups) ×100%. Three separate experiments were performed, with three repeats for each concentration.

#### 2.2.3. Cell Cycle Analysis

To analyze the influence of the chosen compounds on the cell cycle, a flow cytometry analysis using propidium iodide was performed as previously described [37]. The HT-29, MCF7 and HeLaWT cells (3 × 10^5^ cells) were incubated overnight to reach sedimentation and then were treated with 0.5 and 1 × IC_50_ of **KS1**, **KS2**, **KS5**, **KS6**, **KS7** and **KS8**, or Doxorubicin (positive control). Cell cycle analysis was performed after incubation for 24 h. Then, cells were harvested, washed twice with phosphate-buffered saline (PBS), and incubated with a solution of 1% saponin, 50 µg/mL propidium iodide and 10 mg/mL of RNase A (Sigma-Aldrich, Schnelldorf, Germany) in PBS for 1 h at 37 °C. DNA content was analyzed by flow cytometry at the emission wavelength of 488 nm using FACScan (Becton–Dickinson, Franklin Lakes, NJ, USA). The relative proportions of cells in the G1/G0, S and G2/M phases of the cell cycle were determined from the obtained data. Three separate experiments in triplicates were performed for each cell line.

#### 2.2.4. TUNEL Assay

To determine the percentage of apoptotic cells, the TUNEL assay was performed as described elsewhere [38]. TUNEL staining for the detection and quantification of apoptosis at the single-cell level was performed using an in situ cell death detection kit (Roche, Mannheim, Germany) according to the manufacturer’s protocol. Briefly, HT-29, HeLaWT and MCF7 cells (5 × 10^5^/well) were seeded into 6-well plates and cultured overnight before exposure to different concentrations of tested compounds for 24 h. For TUNEL labeling, cells were harvested, fixed, permeabilized and labeled with the TUNEL reaction mixture (enzyme and label solutions) for 1 h at 37 °C in the dark. Negative controls were created by omitting the terminal deoxynucleotidyl enzyme and adding the same volume of the label solution. The cells were analyzed using a FACScan flow cytometer (Becton–Dickinson, Franklin Lakes, NJ, USA).

#### 2.2.5. Statistical Analysis

The obtained data were expressed as the mean ± SD of at least three separate experiments. All statistical analyses were carried out using GraphPad Prism (GraphPad Software, College Station, TX, USA). Differences were assessed for statistical significance using repeated-measures ANOVA. For the MTT assay the threshold for significance was defined as *p <* 0.05, the symbols *, #, ♦, ◦, •, ∇ were used for *p* ≤ 0.05, and the symbols **, ##, ♦♦, ◦◦, ••, ∇∇ for *p* ≤ 0.01. The IC_50_ values were obtained using the nonlinear regression program CompuSyn software version 2022 (ComboSyn Inc., Parammus, NJ, USA) and are presented as estimate ± SE. The viability assessment was calculated using Excel software (Microsoft, St. Louis, MO, USA).

For cell cycle analysis, the threshold for significance was *p* < 0.05, the symbols *, ** were used for *p* < 0.05, and *p* < 0.005, respectively.

For the TUNEL assay, the threshold for significance was *p* < 0.05, the symbols *, **, *** were used for *p* < 0.05, *p* < 0.005, and *p* < 0.001, respectively.

### 2.3. Computational Details

The optimization of the ligands was performed using the Gaussian 16 C.01 program [39] and density functional theory (DFT) formalism with the B3LYP/6–31G(d,p) (very tight criteria) [40] approximation. The crystal structure of the human I kappa B kinase beta (asymmetric dimer; PDB entry: 4kik.pdb), CDK4 in complex with a D-type cyclin (PDB entry: 2w96.pdb [41,42,43]), human ERα LBD (PDB entries: 3dt3.pdb or 1l2j.pdb for ERα or ERβ, respectively) [41,42,44] with the resolution: 2.3000 (2w96.pdb), 2.8300 (4kik.pdb), 2.400 Å (ERα) or 2.950 Å (ERβ) were selected as the biological targets as one of the most used for docking PDB versions of the investigated proteins. To carry out docking simulations (using the AutoDock Vina package [45]), a grid box was defined: 4kik.pdb protein: center_x = 4.584, center_y = 2.443, center_z = 55.258; 2w96.pdb protein: center_x = 4.584, center_y = 2.443, center_z = 55.258; ERβ: center _x = 31.926, center_y = 82.682, center_z = −11.054; ERα: center_x = 41.526, center_y = 1.476, center_z = 15.981; ERβ: center _x = 31.926, center_y = 82.682, center_z = −11.054. In the docking procedure, we considered a distance of d ≤ 4.0 Å between a proton and a heteroatom of the adjacent molecule. The outputs (Appendix A) after the docking procedure (the projections of the 1st poses) were visualized using the LigPlot + v.2.2 software (EMBL-EBI) [46,47].

## 3. Results and Discussion

### 3.1. Chemistry

In this study, the synthesis of the hybrids began with obtaining compounds: mono-oleanoyl hydrogen succinate (**KS1**) and mono-oleanoyl hydrogen succinate methyl ester (**KS1a**), meaning the reaction oleanolic acid or oleanolic methyl ester with succinic anhydride in the presence of 4-(dimethylamino)pyridine (DMAP). The use of a succinyl linker has been successfully applied to the synthesis of biologically active triterpene carboxylic acid derivatives [35]. The presence of the C-28 carboxyl group within the molecule of triterpene increases the polarity of such a derivative, but at the same time decreases the solubility in organic solvents, and in those applied in cytotoxic tests. For this reason, we decided to obtain a group of compounds with esterified carboxyl functions, which make them generally more soluble in many organic solvents than compounds with free carboxyl. To obtain the esters derivatives, the carboxylic group of oleanolic acid in position C-28 was converted into the methyl ester by use of dimethyl sulfate in an ethanolic solution of sodium hydroxide as a methylation agent (Figure 1) [22].

Earlier reports established the improved pharmacological activity of curcumin when its 1,3-dicarbonyl moiety was replaced by isosteric isoxazoles and pyrazoles [48]. In a structure-activity relationship analysis of the β-diketone chain of curcumin, it was postulated that the introduction of a hydrazine fragment to curcumin leads to derivatives where the central 1,3-diketo-enol is masked or made rigid, which in turn improves the anti-tumor activity. The literature also pointed out that the incorporation of the pyrazole ring enhances the antiproliferative activity [49]. The heterocyclic pyrazole and oxazole derivatives of curcumin were obtained using the methods known in the literature [36,50]. Isoxazole (**KS2a**) and pyrazole (**KS2b**) derivatives were prepared by heterocyclization of a diketone system with hydroxylamine hydrochloride and hydrazine dihydrochloride, respectively. Equimolar amounts of curcumin (**KS2**) and hydrazine or hydroxylamine salts were reacted in boiling ethanol solution for 24 h until complete conversion was obtained (Figure 2).

Synthesis of monoesters of curcumin and oleanolic acid has been carried out by a reaction of the curcumin’s phenolic group with a free carboxylic function of a succinyl linker of oleanoyl hydrogen succinate or its C-28 methyl ester. Curcumin (**KS2**) and its heterocyclic analogs (**KS2a**,**KS2b**) were reacted, respectively, with appropriate succinic derivatives of oleanolic acid (**KS1**,**KS1a**) in the presence of dicyclohexylcarbodiimide (DCC) and 4-(dimethylamino)pyridine (DMAP) reagents. DCC is a carboxylic acid activating and coupling agent, while strongly nucleophilic DMAP acts as an acyl transfer in the Steglich esterification reaction, which is a very useful process in converting sterically-demanding substrates. The water resulting from a DCC-mediated reaction reacts immediately with the carboimide applied and forms dicyclohexylurea (DHU), which is insoluble in a reaction medium and precipitates as a white solid. It is, therefore, essential to carry out the reaction in anhydrous conditions [30]. The molar ratio of curcumin or its heterocyclic derivatives, oleanoyl hydrogen succinate or its methyl ester, DCC, and DMAP reagents used in the reaction process are in the sequence 1:1.2:1.6:0.8, and this ratio was optimized during our numerous experiments. Reactions were carried out in an anhydrous chloroform or dioxane solution. The reagents were mixed and then stirred at room temperature for up to 24 h. As a result of synthesis, six hybridized structures with the formulas shown in Figure 3 were produced. Further isolation and purification by column chromatography resulted in the desired pure esters with yields ranging from 51 to 62%.

The structure of hybrid compounds **KS3**–**KS8** was confirmed by ^1^H- and ^13^C-NMR and ESI-MS data. The NMR proton and carbon signals were assigned to the curcumin and oleanolic acid structures, respectively.

The curcumin moiety showed signals for two aromatic rings (7.18–6.90 ppm), two pairs of olefin protons (~7.60 and ~6.50 ppm), the central proton at C4 carbon atom in the seven carbon linker fragment (5.85 ppm for **KS3**/**KS4**, ~6.50 ppm for **KS5**–**KS8**) and two methoxy groups in aromatic rings (3.95–3.69 ppm). Both methoxy groups were observed at different chemical shifts, suggesting that only one phenolic group was esterified with the oleanoyl hydrogen succinate. A broad signal at 15.84/15.87 ppm corresponds to the hydroxyl proton of enolic form observed in the spectrum of compounds **KS3**/**KS4**. ^13^C NMR spectra of hybrids showed signals around 56, 153, 167, 170, 181 and 184 ppm, corresponding to -OCH_3_ (x2), C-NH, C=N, C-OH and C=O, respectively.

The analysis of the spectra of the obtained compounds showed characteristic signals for the oleanolic acid moiety observed as a triplet at ~5.28 ppm (H-12), a multiplet at ~4.50 ppm (H-3), a multiplet in the range of 2.90–2.70 ppm (-CH_2_-CH_2_-), and seven singlets (~1.14, 0.93, 0.92, 0.91, 0.88, 0.80 and 0.76 ppm) for protons from the -CH_3_ groups. ^13^C-NMR spectra of the final compounds contain the groups of signals, typical for the oleanolic acid system, observed at 178–170 ppm (C=O), ~143 ppm (C-13), ~122 ppm (C-12) and ~51 ppm for the carbon of its methyl ester.

### 3.2. Biological Activity

Curcumin has shown considerable anticancer effects against several different types of cancer, including breast [51], cervical [52], colon adenocarcinoma [53,54], prostate [55], colorectal cancer [56], pancreatic [57], and head and neck cancer both [58] in vitro and in vivo. Furthermore, its efficacy and safety in cancer patients, either alone or in combination with other anticancer agents, has been proven in several clinical studies with human subjects [13]. Studies have also shown that oleanolic acid can be used to treat various tumor cell lines, such as MCF7 and MCF7/ADR human breast cancer cells, 1321N1 astrocytoma cell line, hepatocellular carcinoma, HCT-116 colorectal cancer cells and others [59,60]. In addition, OA and its derivatives have been found to be involved in a variety of anticancer signal transduction pathways, including cell proliferation, apoptosis, autophagy, angiogenesis, migration and invasion [60].

#### 3.2.1. Effects of Mono-oleanoyl Hydrogen Succinate, Curcumin, and Their Derivatives on Cells Viability

MTT assay was performed to assess the cytotoxic effect of mono-oleanoyl hydrogen succinate, curcumin and their derivatives on HeLaWT, HT-29 and MCF7 cancer cells, and non-tumorigenic MCF-12A cells. A well-designed cytotoxicity assay should include a non-cancer cell line to check for compound specificity. The MCF-12A cell line was developed from breast tissue with non-malignant fibrocystic disease. It was chosen as the control since it is non-tumorigenic. All of the cell lines were treated with eight compounds (**KS1**–**KS8**) in the concentration range of 0.1–100 µM for 24, 48 and 72 h. Doxorubicin (DOX) was used as a positive control. The obtained results are expressed as a percentage of the untreated cells (Figure 4, Figure 5, Figure 6 and Figure 7).

The half-maximal inhibitory concentrations (IC_50_) of the tested compounds are shown as estimate ± SE in Table 1. In this experiment, the maternal compound **KS1** (mono-oleanoyl hydrogen succinate) in cancer cells decreased the viability of MCF7 cells only, after treatment for 48 and 72 h (IC_50_ = 34.4 and 22.5 µM, respectively). However, the second initial for new hybrids compound **KS2** (curcumin) revealed the highest activity in all of the studied cells, with the highest effect observed in MCF7 cells (15.6 µM for 24 h). Among all of the cancer cells, the activity of compounds corresponding to the IC_50_ value also revealed new hybrids of curcumin and mono-oleanoyl hydrogen succinate, **KS5**, **KS6** and **KS8** compounds. The **KS5** and **KS8** were the most effective in HT-29 cells (IC_50_ = 29 µM and 20.6 µM, respectively); however, **KS6** was the most active in MCF7 breast cancer cells (69.1 µM for 24 h). Compounds **KS3**, **KS4** and **KS7** did not decrease the viability of all studied cells in the applied concentration range (IC_50_ > 100 µM) (Table 1).

All of the studied compounds in non-cancerous breast MCF-12A cells revealed weaker activity than in the MCF7 breast cancer cell line. Moreover, the viability of this cell line was decreased after the treatment of cells with **KS2**, **KS5** and **KS8** for 24 h, and also after cells treatment with **KS1** and **KS6** in longer terms (48 and 72 h) (Figure 7). Compounds **KS3**, **KS4** and **KS7** did not show any effect on studied non-cancerous cells in the applied concentration range (IC_50_ > 100 µM).

#### 3.2.2. The Effect of Studied Compounds on Cell Cycle in Cancer Cells

To verify if the observation in the MTT assay of the decrease of the viability of cancer cells after treatment of cells with **KS2**, **KS5**, **KS6** and **KS8** is linked with the cytostatic or cytotoxic activity of the studied compounds on cell cycle, the induction of apoptosis was investigated. As **KS1** is a maternal compound for the synthesis of the studied derivatives, its activity was also studied. Because of the weak activity of the studied compounds on non-cancerous MCF-12A cells, the cell cycle analysis was performed only in cancer cells. HelaWT, HT-29 and MCF7 cells were treated with **KS1**, **KS2**, **KS5**, **KS6** and **KS8** in concentrations corresponding to 0.5 × IC_50_, and 1 × IC_50_ values. As a positive apoptosis control, doxorubicin (DOX) was applied (15 µM). The results after 24 h treatment of cells are shown in Figure 8.

Cell cycle analysis showed that in all of the studied cancer cells, only the **KS2** compound revealed a decrease in the G0/G1 phase, and an increase in the G2/M phase, observed in HeLaWT in a concentration corresponding to 0.5 × IC_50_, and in both applied concentrations of the compound in HT-29 and MCF7 cells. Moreover, in HeLaWT cells, **KS2** revealed a decrease in the number of cells in the G0/G1 phase in the concentration of compound corresponding to 1 × IC_50_, with an increase of apoptotic cells (up to 10%). An increase of apoptotic cells was also observed in HT-29 cells (7%) in comparison to the control, untreated cells. The rest of the studied compounds in all of the cancer cell lines did not alter significantly the number of cells in particular phases of the cell cycle.

#### 3.2.3. Verification of Apoptosis Induction by Studied Compounds—TUNEL Assay

To verify the induction of apoptosis in studied cells treated with **KS1**, **KS2**, **KS5**, **KS6** and **KS8** compounds, the TUNEL assay was performed. As a positive apoptosis control, doxorubicin was applied (15 μM). In HeLaWT cells treated with **KS2** in a concentration corresponding to 0.5 × IC_50_ and 1 × IC_50_, the number of apoptotic cells increased (30 and 80%, respectively, ** *p* < 0.005; *** *p* < 0.001). However, in HT-29 cells, a slight increase of apoptosis in cells treated with 1 × IC_50_ of **KS2** compound were observed (10%, non-statistically significant). None of the rest of the compounds revealed proapoptotic activity in studied cancer cells (Figure 9).

### 3.3. Computational Analysis

Regarding the effect of the studied compounds on HelaWT cells and the human I kappa B kinase beta (4kik.pdb protein, IKKβ [61,62,63]), the estimates during the docking protocol binding affinities were as follows: −8.200, −9.400, −8.600, −10.100, −8.400, −9.900, −9.700, −9.700, and −8.800 kcal mol^−1^ for **KS1**, keto form of **KS2** (**KS2_keto**), enolic form of **KS2** (**KS2_enol**), **KS3**, **KS4**, **KS5**, **KS6**, **KS7** and **KS8**, respectively. We noticed that fitting the first poses of docked compounds **KS1**–**KS8** using protein 4kik.pdb resulted in forming several hydrogen bonds between the ligands and the protein amino acids (Appendix A). However, the docked poses of the **KS3**, **KS4** and **KS7** analogs were oriented quite similarly (Figure 10). It turned out that the enolic form of curcumin (**KS2_enol** in Figure 10) was slightly more potent in comparison with its keto isomer from the standpoint of binding affinity. Moreover, the **KS5**, **KS6** and **KS8** derivatives were able to form hydrogen bonds with several amino acids within the protein cavity (in comparison with the **KS1**, **KS3** and **KS4**), i.e., Lys147 (**KS8**/3.040 Å), Arg220 (**KS6**/3.180 Å), Arg427 (**KS6**/2.980 and 3.120 Å), Arg579 (**KS6**/3.060 Å) and Arg582 (**KS5**/2.820 Å; **KS6**/3.020 Å).

Considering the activity of the investigated derivatives against the HT-29 cell lines and their ability to interact with the human cyclin D1-cyclin-dependent kinase 4 CDK4 (2w96.pdb protein [39,40,41]), the estimated binding affinities were as follows: −9.00, −7.700, −7.900, −10.000, −7.200, −8.200, −9.900, −7.500, and −8.300 kcal mol^−1^ for: **KS1**, keto form of **KS2**, enolic form of **KS2**, **KS3**, **KS4**, **KS5**, **KS6**, **KS7** and **KS8**, respectively. We observed that the main core of **KS3**, **KS4** and **KS7** compounds was docked in a similar manner (Figure 11). Derivatives **KS1**, **KS3**, **KS4** and **KS7** were able to form only weak hydrogen bonds within the cavity (Appendix A), especially with: Arg126 (**KS1**/3.040 Å), Val27 (**KS3**/3.120 Å), Lys33 (**KS4**/3.100 Å), Arg26 (**KS7**/2.890 Å), Arg126 (**KS7**/3.140 Å) and Gln291 (**KS7**/2.950 Å). For active analogs, more ligand−amino acid contacts were detected, including Lys33 (**KS5**/2.970 Å, **KS6**/3.090 Å), Phe66 (**KS8**/2.790 Å), Glu67 (**KS8**/2.790 and 2.830 Å), Arg126 (**KS5**/2.860 Å, **KS6**/2.810 and 3.010 Å), Asp129 (**KS5**/3.080 Å) and Gln291 (**KS8**/3.110 Å).

The binding of a ligand to the estrogen receptor (ER) is crucial for its potential to act as an ER agonist or antagonist. Curcumin is a compound with an impact on the expression of ERα and p53 in the presence of hormones and anti-hormones in breast cancer cells [64]. On account of the activity of the testes analytes in the MCF7 cell line, in this study, we used in silico techniques to investigate the activity of curcumin analogs to modulate the activity of estrogen receptors using ERα (3dt3.pdb [65]) and ERβ (1l2j.pdb [44]) proteins [66].

Regarding the ERα, the computed binding affinities were as follows: −8.500, −8.00, −7.800, −7.500, −8.200, −9.100, −8.700, −9.800 and −8.900 kcal mol^−1^ for **KS1**, keto form of **KS2**, enolic form of **KS2**, **KS3**, **KS4**, **KS5**, **KS6**, **KS7** and **KS8**, respectively. We observed that the main core of **KS3**, **KS4** and **KS7** compounds was docked in a similar manner (Figure 12). Moreover, the overlap of aromatic rings of the first poses within the active **KS6** and **KS8** derivatives was also significant. Docking protocol allowed us to detect several hydrogen bonds formed between analyzed ligands and amino acids within the ERα, such as (Appendix A): Thr347 (**KS2**_enol/3.140 Å, **KS6**/3.130 Å, **KS8**/2.950 Å), Leu387 (**KS2_keto**/2.740 Å, **KS7**/3.190 Å), Asn519 (**KS4**/3.230 Å), Tyr526 (**KS2_keto**/2.720 Å, **KS2_enol**/2.930 Å), Asn532 (**KS1**/2.850 Å) and Leu536 (**KS7**/3.270 Å). It is known that tryptophan present at position 383 is considered as a conservative point in the hormone binding site, and it is also present in other steroid receptors [66]. Notably, the aromatic rings of Trp383 and **KS6** and **KS8** (distance equaled ca. 4 Å) were coplanar. In case of ERβ, the binding affinities of curcumin analogs were as follows: −7.300, −7.700, −7.800, −7.00, −8.500, −8.00, −7.200, −7.100 and −6.800 kcal mol^−1^ for **KS1**, keto form of **KS2**, enolic form of **KS2**, **KS3**, **KS4**, **KS5**, **KS6**, **KS7** and **KS8**, respectively. We also observed that the binding modes of ligands **KS4** and **KS7** were similar (Figure 13). Hydrogen bonds within the ligand-protein complex (Appendix A) were not detected (**KS7**) or were weak (HBs formed with Lys395 for **KS3**/3.280 Å, with His350 for **KS4**/3.200 Å). Several hydrogen bonds formed between other analyzed ligands and amino acids within the ERβ, such as (Appendix A): Glu276 (**KS1**/3.010 Å), Pro277 (**KS2_enol**/2.960 Å), His279 (**KS2_enol**/2.960 and 3.010 Å), Glu305 (**KS2_keto**/2.960 Å, **KS2_enol**/3.110 Å, **KS5**/2.810 Å, **KS6**/2.920 Å), Trp345 (**KS8**/3.260 Å), Arg346 (**KS1**/2.880 Å), Gln393 (**KS8**/3.050 Å), His394 (**KS1**/3.260 Å) and Tyr397 (**KS2_keto**/2.960 Å).

### 3.4. Structure-Activity Relationships

Comparing the cytotoxicity results in Table 1, mono-oleanoyl hydrogen succinate (**KS1**) was inactive against HeLaWT and HT-29 cells, but quite active against MCF7 cells. The conjugation with curcumin, which was active against all tested lines, resulted in a significant decrease in cytotoxic activity, with both curcumin oleanoyl hydrogen succinate derivative (**KS4**) and its methyl ester (**KS3**).

It was also noted that the conversion of the keto-enol moiety in compounds **KS3** and **KS4** to the corresponding pyrazole **KS8** and **KS6** leads to increased cytotoxicity against all cell lines. However, in the case of isoxazole derivatives hybrids (**KS3** and **KS4)**, only the compound with the free carboxylic group in the oleanolic acid skeleton (**KS5**) showed a significant increase in activity. Thus, the ring substituents affected the increased cytotoxicity in the pyrazole and isoxazole derivatives obtained. The transformation of the free carboxyl function of the oleanolic acid skeleton in **KS6** into methyl ester (**KS8**) increased the cytotoxic activity of the **KS6**. Only curcumin isoxazole mono-oleanoyl hydrogen succinate methyl ester (**KS7**) was inactive.

In conclusion, curcumin was the most potent against all tested cell lines, but several of the obtained hybrid compounds, in which the one hydroxyl group was linked with an oleanolic acid skeleton, showed significant cytotoxicity against HeLaWT, HT-29 and MCF7 cells.

The current direction in the field of discovering new drugs in the treatment of breast cancer is testing currently used drugs—directed to the treatment of specific subtypes of breast cancer—in combination with new ones with proven activity, or the use of combinations of various substances/compounds with biological activity to enhance this effect. Our research, although aimed at verifying such a trend in the context of increasing therapeutic effectiveness, did not show the expected spectacular effect, but is important for the design of new drugs, indicating a different type of modification than the examples of hybrids presented by us.

## 4. Conclusions

In this study, hybrid compounds were synthesized through the combination of curcumin and oleanolic acid skeleton; novel promising curcumin-triterpene analogs containing heterocyclic rings in curcumin part were also synthesized. The potential cytotoxic effect of these newly synthesized agents was investigated in an experimental in vitro model.

The tested compounds **KS2**, **KS5**, **KS6** and **KS8** showed cytotoxic activity against HeLaWT, HT-29 and MCF7 cells, and **KS2** was the most active in MCF7 cells. Curcumin showed the strongest activity, followed by compound **KS8**, then compounds **KS6** and **KS5**. The cytotoxic activity of **KS1** against MCF7 cells was at the level of that of **KS8**.

Based on in silico results, we can assume that computations proved the low activity of the **KS3**, **KS4** and **KS7** analogs towards IKKβ, CDK4 proteins, ERα and Erβ, basically due to their similar binding modes and the weak nature of hydrogen bonds formed within the cavity of ligand−protein complexes.

Overall, the conjugation of the structure of curcumin and oleanolic acid may represent an important basis for the development of effective anticancer agents, without retaining the diketone moiety. Moreover, some modifications in this portion have led to obtaining more information about different biological activities on several cancer cell lines.

However, our research shows that not every combination of chemical modifications of the structure of two biologically active compounds is effective in the context of anticancer activity. This requires a multi-directional view; modifications that would increase the solubility, bioavailability and selectivity of a new hybrid are not always appropriate. Therefore, the research results presented by us only indicate the direction of designing new drugs, with an emphasis on specific modifications and combinations of two biologically active compounds in order to improve the effectiveness of their biological action and create an effective and selective anti-cancer drug.

## Data Availability

Not applicable.

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
