# Peer review of "Synthesis, Cytotoxicity and Molecular Docking of New Hybrid Compounds by Combination of Curcumin with Oleanolic Acid"

_biomedicines, 2023, doi:10.3390/biomedicines11061506_

Round 1

Reviewer 1 Report

It is interesting to read the manuscript titled New Hybrid Compounds Based on Curcumin and Oleanolic Acid: Synthesis, Cytotoxicity and Molecular Docking” by Sowa-Kasprzak et al. I appreciate the authors' efforts in conducting the study, which will be more beneficial to scholars working in the new drug discovery for the treatment of cancer. However, if the following changes are made and they are incorporated into the manuscript, it could be considered for publication in Biomedicines.

 1.     I recommend the author to change the title as “Synthesis, Cytotoxicity and Molecular Docking of New Hybrid Compounds by Combination of Curcumin with Oleanolic Acid” which is more appropriate

2.     Line 15, The word “In the article” to be changed as “In the present study”

3.     was unable to locate the cytotoxicity results in the abstract. Additionally, the abstract's conclusion section has to be expanded to include with future perspective.

4. Lines 41-44 should be supported by the most recent reference (https://www.ncbi.nlm.nih.gov/pmc/articles/PMC8990857/).

5.     Mention C-3, C-12 and C-28 positions in the structure of curcumin in Figure 1

6.     Double-check and maintain the following term hours/h in the manuscript for consistency. Apart from that, the methodology section is well written with necessary references.

7.     In the cytotoxicity study, the authors must describe the exact cytotoxic effects of curcumin and oleanolic acid on the cancer cells that were used. Afterwards can the authors compare the results of the novel compounds to those of curcumin and oleanolic acid alone.

 8.     The authors also did a good job at writing the results section. The manuscript does an excellent job of illustrating the Tables and Figures. The discussion's lack of explanation, nonetheless, caught my attention. In order to make it clear how the current research findings fit into the larger context of what is currently being done regarding new drug discovery for the treatment of breast cancer. I would advise the authors to refocus their discussion rather than incorporating more background information about the literature.

 9.     I also propose authors to provide a critical rationale for their findings in the conclusion. Therefore, everyone will see the importance of this research. The conclusion must also address potential points of view. The author should stress the significance of this research.

It is a well-written manuscript. However, minor English language correction is necessary.

Author Response

First, we would like to thank you for your kind comments and constructive criticism. All comments taken into account are included in the attached document. All changes are highlighted in different colors.

Reviewer 2 Report

Sowa-Kasprzak et al. reported the anti-cancer activity of hybrid compounds based on Curcumin and Oleanolic Acid. The study is interesting, and the authors discovered a new class of hybrid molecules with effective anti-cancer activity. I have some major concerns which need to be addressed before publication.

1.      Hydrogen and carbon NMR spectra should be provided in the supplementary file.

2.      The authors did not check the purity of these hybrid molecules; hence authors should provide HPLC purity for these compounds.

3.      Authors should also check apoptosis-associated biomarkers by western blot to find out the mechanism of action of these new anti-cancer agents.

4.      The physicochemical and Pharmackonitic profile of these hybrid molecules should also be investigated.

5.      Typo and grammatical errors should be corrected. 

  Typo and grammatical errors should be corrected. 

Author Response

(The authors gave the same response as above.)

Reviewer 3 Report

Dear Authors

The manuscript (biomedicines- 2350219) entitled “New Hybrid Compounds Based on Curcumin and Oleanolic Acid: Synthesis, Cytotoxicity and Molecular Docking” is well written, has an important scientific message and should be of great interest to the readers of Biomedicines journal (ISSN 1999-4923) (Section: Drug discovery, Special issue: State-of-the-Art Drug Discovery and Development in Poland).

The manuscript presents interesting and scientific important results. The authors discuss how Curcumin and oleanolic acid are used to develop bioactive hybrid compounds that are intended to target cancer cells. They have studied the cytotoxicity against different cancer cell lines. Additionally, in silico studies were also performed. The authors concluded that curcumin-triterpene hybrids represent an important class of compounds for the development of effective anticancer agents also without the diketone moiety in the curcumin molecule. These findings may be of interest for further research into combinations of curcumin and oleanolic acid derivatives.

- The work is original and contains new results that significantly advance the research field of phytochemistry, computational chemistry and molecular docking, pharmacology of natural products and anticancer activity. The article contains material that is new or adds significantly to knowledge already published.

- The results are interesting and important to researchers and scientists in relevant fields of in vitro and in silico anticancer effect of natural bioactive compounds, and pharmacognosy.

- Sufficient references are cited for providing a background to the research.

- The overall structure of the article is well organized and well balanced. The article is written with the minimum length necessary for all relevant information.

- The conclusion is logically supported by the obtained results.

However, few issues need to be resolved before the final acceptation of the paper.

Line 18

Their cytotoxicity against different cancer cell lines was evaluated and compared with the activity of curcumin.

--- Authors must insert some results (median inhibitory concentration IC50) related to the cytotoxicity of curcumin and derivatives against different cancer cell lines (HELA breast and colorectal adenocarcinoma).

Lines 20-21

The computations determined the activity of the tested compounds towards proteins selected due to their similar binding modes and the nature of hydrogen bonds formed within the cavity of ligand−protein complexes.

---- It is important to give more details regarding the finding of molecular docking of ligands with different protein receptors, such as the binding energies, and the nature of amino acids involved in the active site interactions.

INTRODUCTION SECTION

The Introduction that does not establish the background of the problem studied = Insufficient explanation of the rationale for the study, and insufficient literature review            

--- It is very important for authors to present a persuasive and rational argument in their papers. You should be able to convince readers that your research is both sound and important through your writing. 

Line 31

The contemporary strategy to fight cancer is a consequence of the rapid development of knowledge about the molecular basis of cancerous diseases.

--- Authors must insert some references after each sentence or paragraph.

Line 33

They inhibit cell division, cause DNA damage, block transcription and translation, and contribute to the induction of tumor cell 34 apoptosis in the final stage.

--- What about the role of antiangiogenic effect? It has been reported that angiogenesis inhibitors are unique cancer-fighting agents because they block the growth of blood vessels that support tumor growth rather than blocking the growth of tumor cells themselves. Angiogenesis inhibitors interfere in several ways with various steps in blood vessel growth.

Line 35

The resistance of tumor cells to drugs forces scientists to constantly search for new substances and modify existing compounds to use them in targeted cancer therapies.

--- The drug resistance is not only the cause of searching new anticancer compounds. Side effects that occur during chemotherapy treatment has been reported in several anticancer drugs. Permanent side effects of chemotherapy may include damage to your heart, lungs, kidneys, nerve endings or reproductive organs. It is important to emphasis this aspect.

Line 39

The use of phytochemicals is highly desirable because of the large availability and readiness of use, low cost and the use of low cytotoxicity relative to normal cells [1].

--- In my opinion, I do not agree with this statement. It has reported that the potential toxicity of traditional herbal medicines can arise from acute or chronic exposure even with extracts of low toxicity. Some plants well known in traditional medicine to be toxic or poisonous. Safety training is essential when working in a plant.

Page 2: Line 1

A wide variety of cellular properties of curcumin have been demonstrated, including antiproliferative [3], pro-apoptotic, anticancer [4], antioxidant, anti-inflammatory [5] and anti-bacterial activities [6].

--- Please you need to give more details (in vitro or in vivo) about these biological activities and the findings obtained through these studies.

Line 7

Apart from its ecological roles in plants, some pharmacological activities such as anti-tumour, anti-inflammatory, anti-oxidant, anti-diabetic and anti-microbial effects have been attributed to oleanolic acid in different models of diseases [8]

--- Same comment as previously reported. Authors must give more details about these pharmacological properties and findings.

Figure 1. Modifications possibility of the chemical structure of curcumin and oleanolic acid

--- Please give the explanation related to the yellow circles inserted in Figure 1.

Page 3; Line 6

Figure 2. General formula of hybrids derived from curcumin and oleanolic acid

--- Same comment as reported previously in Figure 2.

Page 6; Line 26

All cell lines were grown near confluence in 100x15 mm cell culture Falcon® Petri dishes (Corning, Poland) at 37°C in an atmosphere containing 5% (v/v) of CO2 at 95% (v/v) of relative humidity.

--- What about the antibiotic and antifungal used in the cell culture technique?

Line 31

Cytotoxicity of studied compounds was assessed on a broad spectrum of mentioned cell lines using the MTT test.

--- What about the positive control (chemotherapy drug) used for the MTT assay to assess the cytotoxic activity of curcumine and analogues?

Line 45

The IC50 values were obtained using the nonlinear regression program CampuSyn software version 2022 (ComboSyn Inc., Parammus, NJ, USA) and are presented as estimate± SE. The viability assessment was calculated using Excel software (Microsoft, USA).

--- I think that it would be appropriate to insert this sentence in the section of “2.2.3. Statistical analysi”.

Line 45

The outputs (Figure S1-S4) after the docking procedure (the projections of the 1st poses) were visualized using the LigPlot + v.2.2 software (EMBL-EBI) [30][31].

--- I do not find the interactions images generated by LigPlot software in the manuscript. Please check your images in the whole manuscript.

--- authors missed an important point related to the ADMET properties of curcumin and derivatives. The ADMET (absorption, distribution, metabolism, excretion, and toxicity) properties of your molecules are of vital importance. For example, authors could use ADMETlab for the predictions of 7 basic physicochemical properties, 28 ADMET-related properties and 6 drug-likeness.

Figure 4

Viability assessment of HeLaWT cells after treatment with mono-oleanoyl hydrogen suc- 16 cinate, curcumin, and their derivatives for 24, 48, and 72 h.

--- It is necessary to insert the statistical differences between all tested compounds (KS1 to KS8) by ANOVA test followed by a post hoc comparison test.

--- Same comment with Figure 5, 6 and 7.

Application of statistics in the methodology and results sections of a manuscript creates an extra edge over the others, statistics being the need of the moment. Precisely showing the results with application of statistical principles will increase the probability of acceptance of the manuscript.

Line 10

Computational Analysis

--- Please insert a table of the molecular docking results which report all amino acids involved in the binding interactions, distances and type of interactions.

The authors should have a sufficient know-how to interpret the exact reasons of the research outcome. Even if the results are out of specifications, the author should be able to critically interpret the cause in the discussion section. It is not mandatory to show positive outcomes alone. Manuscripts can support future research if they accurately interpret the root cause of the negative results.

Final decision: Accept after major revisions.

Author Response

(The authors gave the same response as above.)

Round 2

Reviewer 1 Report

The author responded to all of my inquiries and revised the manuscript content accordingly. As a result, I suggest that it be taken into consideration for publication in Biomedicines in its present form.

Minor editing of English language required.

Reviewer 2 Report

The authors have revised the manuscript as suggested. The current version of ms may be accepted for publication.

Minor editing of English language required

Reviewer 3 Report

Final decision : Accept in present form